# Coping Styles among People with Parkinson’s Disease: A Three-Year Follow-Up Study

**DOI:** 10.3390/bs10120190

**Published:** 2020-12-12

**Authors:** Maria H. Nilsson, Frank Oswald, Sebastian Palmqvist, Björn Slaug

**Affiliations:** 1Department of Health Sciences, Faculty of Medicine, Lund University, 22100 Lund, Sweden; maria.h.nilsson@skane.se; 2Memory Clinic, Skåne University Hospital, 20502 Malmö, Sweden; sebastian.palmqvist@skane.se; 3Clinical Memory Research Unit, Faculty of Medicine, Lund University, 22100 Lund, Sweden; 4Interdisciplinary Ageing Research (IAW), Goethe University Frankfurt, 60323 Frankfurt am Main, Germany; oswald@em.uni-frankfurt.de

**Keywords:** adaptation, coping, chronic illness, parkinson’s disease, psychosocial adjustment

## Abstract

People with Parkinson’s disease (PD) experience a gradual loss of functional abilities that affects all facets of their daily life. There is a lack of longitudinal studies on coping styles in relation to the disease progression among people with PD. The aim of this study was to explore how coping styles in PD evolve over a 3-year period. Data from the longitudinal project “Home and Health in People Ageing with PD” was utilized (N = 158), including baseline and 3-year follow-up assessments. Coping was captured by ratings of 13 different coping styles. A factor analysis was conducted to analyse patterns of coping styles. Stability and change were analysed for each of the 13 styles with respect to the course of the disease. The factor analysis revealed four coping patterns: pessimistic, optimistic, persistent and support-seeking. The stability of each coping style over time ranged from 75.3% to 90.5%. Those who experienced a worsening of the disease were most inclined to change their coping style (*p* = 0.006). The results suggest that even when facing severe challenges due to PD in daily life, coping styles remain relatively stable over time. However, a worsening in PD severity appeared to trigger a certain re-evaluation of coping styles.

## 1. Introduction

Living with a progressive and chronic neurological disease such as Parkinson’s disease (PD) imposes challenges in daily life. PD is characterized by both motor-(bradykinesia, tremor, rigidity and postural instability) and non-motor symptoms such as decreased cognitive functioning, depressive symptoms and fatigue [1,2]. Thus, as the disease progresses, people with PD may experience a gradual increase in how the disease influences their ability to maintain daily life as they know it [3]. Several studies have investigated the consequences and impact of living with PD on daily life [4,5,6], and people with PD have expressed the need to adapt to changes and losses, such as functional abilities, work, social and role functioning [7,8], in order to maintain quality of life and continue living as normally as possible [9]. A characteristic aspect of living with PD is the unpredictability of the manifestations of the disease and the gradual loss of control over the body [10]. Common strategies for managing everyday life that are described in the literature include active information seeking [6,11], planning and structuring [12], trying to have a positive attitude/outlook [13,14] and being physically active [15] as well as utilizing social support [16].

The coping strategies embraced by an individual may depend on different factors, such as personality, progression of the disease and severity of challenges faced in daily life [17,18]. With regard to the general population, the psychological literature on coping with age-related health decline has suggested that some persons tend to be more accepting and let matters take their own course, that is, showing more “regressive” coping strategies. In contrast, other persons show more “mature” coping strategies and approach their disease in an active and problem-solving manner [19]. Though much of the research on coping with PD have focused on similar or related distinctions, especially as between passive and active coping [20,21], there are also studies that have shown that individuals may use a pattern of different coping strategies as a repertoire of useful behaviors in different situations [22,23]. Coping strategies that are consistently displayed in different situations and over time are also known as “coping styles” [24]. Longitudinal studies on stability and change in coping in later life in general have revealed the need to consider not only functional strategies but also coherent coping styles [19,25]. Research findings from both cross-sectional and longitudinal studies on coping with PD have also provided evidence on the need to address patterns as well as overlapping and mixed styles of coping [23,26].

Still, knowledge on stability and change in coping among persons with PD is limited, and differentiated data on change in coping with respect to different courses of disease over time are largely lacking. To the best of our knowledge, only four longitudinal or follow-up studies have explicitly addressed coping in people with PD. In one earlier study (N = 70), coping patterns were examined over time (2-year follow-up) in light of two theoretical approaches to coping, a contextual and a dispositional approach [22]. With the contextual approach, coping is expected to change in response to changes in the disease, whereas a dispositional approach considers coping as a component of the personality that is relatively stable over time. The findings of the study by Frazier largely supported the dispositional approach, as coping was found to be rather stable from baseline to follow-up. However, there was also some evidence of the validity of a contextual approach, as change in coping style was significantly correlated with less stress and higher quality of life. Another study (N = 75) investigated the role of coping and social support for quality of life or people with PD at two measurement points with one year in between [27]. The latter study showed that quality of life was relatively stable from baseline to the 1-year follow-up, and there was only a weak relation between active coping strategies and quality of life. The two other longitudinal studies include a psychometric study of a self-report instrument for measuring coping with stressful situations [23] tested in a PD population, and a study of the dynamics of illness representations and coping strategies for PD patients that underwent Deep Brain Stimulation [28]. Consequently, there is a need for further research to explore stability and change in coping styles in PD with larger samples that are more differentiated with respect to the dynamic of the illness during longer follow-up periods.

The overall aim of this study was to explore how coping styles in PD evolve over a 3-year period. More specific aims were to (1) analyse patterns in coping styles, and (2) examine stability and change of coping styles, specifically in relation to different courses of disease, i.e., those who progressed into more severe stages of PD versus those who remained relatively stable in low or high stages of PD severity over time.

## 2. Materials and Methods

The present study utilized data from a large longitudinal Swedish research project; “Home and Health in People Ageing with PD” (HHPD). This study includes baseline assessments (2013) and a 3-year follow-up (2016). Details of the project design (including sample size calculations) are published in a study protocol [29].

### 2.1. Recruitment and Study Participants

At baseline, a sample of 653 participants (recruited from three hospitals in Skåne, Sweden) met the inclusion criterion for HHPD of being diagnosed with PD (G20.9, according to ICD-10) at least one year before inclusion in the project. Out of these, 216 individuals were excluded according to the following exclusion criteria: difficulties in understanding/speaking Swedish (*n* = 10), severe cognitive difficulties (*n* = 91), living outside the present health care region (Skåne; *n* = 58) or other conditions making it difficult for the individual to participate in the study, such as hallucinations, recent stroke etc (*n* = 57). That is, a potential participant was excluded if deemed unable to give informed consent or partake in the majority of the data collection. The remaining 437 individuals were invited to participate but 22 were unreachable, two had their PD diagnosis revised, 157 declined, and one was excluded due to extensive missing data. Of the 255 that participated at the first data collection occasion (T1, i.e., baseline), 165 completed the second collection (T2, i.e., 3-year follow-up). 

For the present study, seven additional participants were excluded due to missing data on coping styles. Consequently, the final sample size was 158 (65% men). At T1, the participants mean (SD, minimum–maximum) age was 68.5 (8.6, 46–91) years; the median (q1–q3, minimum–maximum) PD duration at T1 was 8 (5–12, 2–43) years. Participant characteristics are presented in Table 1.

The HHPD project was approved by the Regional Ethical Review Board (Nos. 2012/558; 2015/611) in Lund, Sweden. All participants provided written informed consent. The dataset for the current study is not publicly available due to ongoing longitudinal data collections.

### 2.2. Data Collection Procedure

The data collection included a self-administered postal survey followed by a subsequent home visit that involved interview-administered questions and questionnaires, observations and clinical assessments. The home visits were scheduled at the time of day when the participants reported usually feeling at their best (i.e., being in the “on state”). If the participant became too tired or unwell (i.e., became increasingly “off”), the assessments were postponed and completed on another day at the earliest possible date (which, in this sample, happened in two cases at baseline and in three cases at 3-year follow-up). Conducting the assessments when the participants were “off” would potentially make it difficult for the participants to adequately reflect and respond, which would thus render the data invalid. All data-collectors were health care professionals, and they underwent project-specific training before the data collection was initiated. 

### 2.3. Coping Styles

To capture coping styles in our sample, we utilized the Coping Patterns Schedule (CPS) instrument [19,30]. The CPS includes a set of 13 statements covering more complex and biographically developed ways of dealing with problem situations in everyday life and daily hassles, beyond specific and mere functional strategies. Each statement represents a certain coping style, and the respondent rates how well it fits their own thinking and behavior on a five-point Likert scale (1 = strongly disagree, 2 = disagree, 3 = neutral, 4 = agree, 5 = strongly agree). The instrument is administered as a structured interview. For an overview of the statements and the coping styles they represent, see Appendix A. The CPS instrument was developed and used in the Bonn Gerontological Longitudinal Study [25] and the Berlin Ageing Study (BASE) [19], supporting the validity of the instrument. 

### 2.4. Demographics, Functional Limitations, Cognitive Functioning and PD Severity

Data on demographics included age and sex. Functional limitations (12 items) and dependence on mobility devices (two items) were assessed dichotomously (present = 1/not present = 0) by a trained rater using a combination of interview and observation as part of the Housing Enabler instrument [31]. This included the following 14 items: difficulty interpreting information; visual impairment; blindness; loss of hearing; poor balance; incoordination; limitations of stamina; difficulty in moving head; reduced upper extremity function; reduced fine motor skills; loss of upper extremity function; reduced spine and/or lower extremity function; dependence on walking aids; dependence on wheelchair. Global cognitive functioning was clinically assessed using the Montreal Cognitive Assessment (MoCA, scored 0–30; higher = better) [32]. PD severity was clinically assessed (in “on state”) according to Hoehn and Yahr (HY) [33]. The HY includes five stages: HY I (unilateral involvement, usually with minimal or no functional disability); HY II (bilateral involvement without impairment of balance); HY III (unilateral or bilateral + postural instability); HY IV (severely disabled; still able to walk or stand unassisted); and HY V (confined to bed or wheelchair unless aided). For analyses of coping styles in relation to PD severity, we defined three different HY subgroups with respect to their courses of disease: (1) those that remain at stages HY 1–2 from baseline to follow-up (“Low stable”, *n* = 58), (2) those that decline from HY stages 1–2 at baseline to HY 3 or worse at follow-up (“Worsening”, *n* = 29), and (3) those that are already at HY 3–5 at baseline (“High stable”, *n* = 71). 

### 2.5. Additional Information

Descriptive variables included: education (high-school, university); living area (rural/urban); PD-duration (years); motor symptoms (Unified PD Rating Scale, UPDRS part III, scored 0–108; higher = worse) [34]; total number of non-motor symptoms (Non-Motor Symptoms Questionnaire for PD, NMS Quest, scored 0–30; higher = worse) [35], and freezing of gait (FOG Questionnaire, self-administered version, FOGQsa, scored 0–24; higher = worse) [36,37] and independence in ADL (PD Activities of Daily Living Scale, PADLS, scored 1–5; higher = worse) [38]. Descriptive information of the sample is provided in Table 1.

### 2.6. Statistical Analyses

To analyse patterns in the ratings of different coping styles, we carried out a factor analysis on the Coping Patterns Schedule, using principal component analysis as extraction method, and varimax rotation with Kaiser Normalization [39]. 

To examine if the rating of each coping styles was stable or had changed between T1 and T2, we compared the ratings at the two timepoints and defined a difference ≤1 (i.e., on an individual level) as stable, and a difference ≥2 as a change. To capture stability across all coping styles, we created a variable that reflects the total number of coping styles (minimum–maximum, 0–13) that were stable (i.e., differed ≤1 between T1 and T2).

To compare stability of coping styles, age, number of functional limitations/dependence on mobility devices and MoCA score between the three HY subgroups, we utilized the Kruskal–Wallis test, and Mann–Whitney U-test for ad hoc pairwise tests. To compare the distribution of sex, we used Chi-square test. *p*-values < 0.05 were considered statistically significant. The SAS software (SAS Institute Inc., Cary, NC, USA) version 9.4 was used for all analyses.

## 3. Results

### 3.1. Coping Patterns

The factor analysis revealed a clear four factor solution with no cross-loadings, explaining 50% of the variance and reflecting four coping patterns covering different styles of coping in each pattern. We chose to label the two most prominent factors (i.e., coping patterns) “Remaining faithful but often feeling blue” and “Being even-minded and optimistic despite ups and downs”. They incorporated 10 out of the 13 coping styles and explained 33% of the variance (Table 2).

The pattern “Remaining faithful but often feeling blue” contained six coping styles, which reflected relatively passive and partially pessimistic ways of coping. “Being even-minded and optimistic despite ups and downs” contained four coping styles, which reflected relatively optimistic, though still somewhat ambivalent, feelings and behavior. The third factor reflected a persistent and problem-oriented coping pattern that we labeled “Keep going and adapt to the situation”, including two related coping styles. Finally, the fourth factor included only one prominent coping style, reflecting the social way of coping that we labeled “Getting help and social support”. In short, the four coping patterns could be considered as: pessimistic (Factor 1), optimistic (Factor 2), persistent (Factor 3) and support-seeking (Factor 4). For further details, see Table 2.

### 3.2. Stability of Coping Styles over Time

The proportions of individuals not changing their assessment of each coping style between T1 and T2 (i.e., delta score of ±1), ranged from 90.5% to 75.3% depending on coping style. The three most stable coping styles were “Humor”, “Life loses meaning” and “Faith”. Across all participants, the coping styles that changed most from T1 to T2 in terms of an increase in individual mean rating scores were “Keep going” (3.4 to 3.8), “Ups and downs” (3.4 to 3.7), “Wish for information” (3.7 to 3.9), “Humor” (3.6 to 3.8), “Social support” (3.4 to 3.8), “Someone else to take over” (2.3 to 2.5), and “Giving up” (2.2 to 2.4). Overall, the average differences between T1 and T2 in ratings of each of the 13 coping styles ranged from 0.0 to 0.4 on a scale from 1 to 5. The top styles at T1 in terms of mean rating were “Comparison with the past”, “Adaptation to the given” and “Wish for information”. At T2 the top styles were “Wish for information”, “Keep going”, “Social support”, and “Humor”. For detailed information, see Table 3.

### 3.3. Subgroups with Respect to Different Courses of PD Severity

In order to examine clinically meaningful changes in the coping styles over time, we coupled changes and stability in all 13 coping styles with three different subgroups of clinical disease progressions, instead of just reporting changes and stability in the total sample over time. The three different disease progression groups were, according to changes in the HY scale, (1) “Low stable”, (2) “Worsening”, and (3) “High stable”. With respect to the mere number of stable versus changed coping styles, comparisons between the three groups revealed that participants in “Low stable” were least inclined to change their coping styles (*p* = 0.006), with 12 of 13 styles being stable between T1 and T2. The group who declined in HY score (“Worsening”) was most inclined to change coping style, with changes in rating exceeding one step on the CPS scale for three of the coping styles. The “High stable” group was in between the other two with 11 of 13 styles being stable. Ad-hoc pairwise tests showed that the only significant difference was between the “Low stable” and the “Worsening” group (*p* = 0.002). The “Low stable” group was younger, had significantly less functional limitations and better MoCA scores, both at T1 and T2, compared to the other groups (*p* < 0.001 in all comparisons). There were no differences between the groups with regard to sex. See Table 4 for details.

### 3.4. Coping Styles over Time and in Relation to Different Courses of PD Severity

The top coping styles differed between those who were stable in terms of PD severity, those who experienced a worsening of PD severity and those who were at a severe stage of PD in terms of HY classification already at baseline. For details, see Table 5, Table 6 and Table 7.

The changes in rating in the “Low stable” HY group was mostly within 0.0–0.3 for each coping style. “Wish for information”, “Humor”, “Ups and downs” (all Factor 2), as well as “Keep going” (Factor 3), rated relatively high at T1 and increased to T2. “Social support” and, on a lower level, “Giving up” (Factor 1) increased as well. “Adaptation to the given” (Factor 3) and “Comparison with the past” (Factor 2) decreased slightly over time. Apparently, styles from the optimistic pattern mostly increased over time in this relatively healthy subgroup (Table 5).

In the “Worsening” subgroup with considerable health decline, there was a notable change in ranking and mean rating for coping style “Social support”, which had an increased rating of 0.6 between T1 and T2 and was the most highly rated coping style at T2. Correspondingly, coping style “Comparison with others”, which rated among the top at T1 and was stable in both other subgroups, had a markedly decreased rating of 0.5 from T1 to T2. Thus, whereas internal social comparisons might become less important for coping, active social support might become more important during this health deterioration process (see Table 6).

For the “High stable” subgroup (i.e., severe HY class already at baseline), it was notable that the persistent coping style “Keep going” (belonging to factor 3) was rated already relatively high at T1, but increased significantly to the top ranked style at T2 (+0.5). Other coping styles seemed to remain stable in either ranking and/or mean rating. This might reflect the specific role of proactively addressing one’s own situation (Factor 3) in a relatively stable but unhealthy situation (see Table 7).

## 4. Discussion

The main findings of the present study were two-fold. First, the four distinct coping patterns identified (pessimistic, optimistic, persistent and support-seeking) and the relative stability of coping styles over a 3-year period showed coping-related difference with respect to the group of persons with PD in general. Second, with respect to a more differentiated view on several courses of the disease, changes and stability in coping styles within subgroups of persons with PD could be detected and interpreted towards the existing literature and consequences for intervention. Specifically, a marked disease progression was associated with a stronger inclination for change in coping, whereas mild and stable PD was least associated with inclination for change.

### 4.1. Coping Patterns

With respect to coping with PD in general, the four factors or patterns of coping styles revealed by the present study underpin previous research on the complexity and ambivalence of coping among persons with PD [22]. The two most prominent patterns detected, the mostly pessimistic “Remaining faithful but often feeling blue” and the mostly optimistic “Being even-minded and optimistic despite ups and downs” could also be compared to the more common distinction between passive and active coping styles [20,21]. However, the current results also suggest that there is some mix of them as the content in each of the patterns is not purely passive or purely active. This is in line with evidence of overlap between passive and active styles found by Liebermann and colleagues [26] in a recent study. These two coping patterns also resemble findings of an early study by Dakof and Mendelsohn [40] where two of four clusters emerging from analysis of interview data with PD patients were identified as “Passive and resigned” and “Sanguine and engaged”, respectively. Despite reasons to question the general categorization of coping styles in terms of active and passive, one might nevertheless argue that the findings of the current study support the concept of two major forms of coping styles, which is also reflected in previous research [20,21,25]. A result that stood out was that “Wish for information” was such a prominent coping style at both baseline and the 3-year follow-up, regardless of whether the PD severity had changed or not. This finding is in line with several previous PD-studies that described the need for information and suggests that active information-seeking is an important strategy [6,11,41,42].

### 4.2. Stability of Coping Styles over Time

Three years after baseline, the role of the different coping styles remained largely the same at first glance, though there was a range (from 75% to 90%) suggesting that some coping styles changed less than others. Using “Humor” as a coping style is the most stable of all, and is arguably a disposition that is deeply embedded in the personality [43]. These findings thus seem to support a dispositional theory of coping and could be interpreted in line with the only previous longitudinal study that has examined coping in relation to progression of PD [23]. According to this theory, a repertoire of coping strategies is developed and consolidated through early life and, like personality, it remains relatively stable from adulthood and onwards [24], although this would not hold for an empirical proof from a more differentiated perspective. Thus, there is evidence in the literature that if confronted with a new situation, such as the progression of a chronic disease, coping strategies that do not appear helpful any longer may be changed [22].

### 4.3. Subgroups with Respect to Different Courses of PD Severity

With respect to coping among subgroups of persons with PD, the results of the present study give some support to such a contextual theory of coping. According to this theory, the use of particular coping strategies is determined by contextual elements, such as duration and stage of a disease, but also by the perceived degree of helpfulness of previous attempts of coping. To explore changes and stability in coping styles in relation to courses of the disease, in this study we used a combination of mean rating of styles and number of stable styles to examine what happened among members of the three groups between T1 and T2. Changes were applied especially in the “Worsening” HY subgroup, while the stable HY subgroups were less inclined to obvious change. Nevertheless, styles from the optimistic pattern seem to considerably increase over time in the healthy subgroup, and proactively addressing one’s own situation (“Keep going”) seems to increase in the stable but unhealthy group. However, with respect to the content of the most notable changes in the “Worsening” group, the style “Comparison with others” decreased and seeking for “Social support” increased over the period of three years. This may reflect that internally processed social comparisons might become less helpful when facing health deterioration, whereas real social support might be needed to cope with the situation. Thus, social support emerges as an important coping strategy in the literature on coping and PD [44,45,46], as well as in this study. It is a noteworthy result of the present study that “Social support” appears at the other end of the stability ranking list, indicating that it is more likely to be chosen by a person if it is expected to be helpful, even if it has not been considered as a strategy previously in life. Prior studies have described that social support facilitates participation among people with PD, i.e., both in relation to activities and exercise; family, friends, people in the workplace and peers can act as external motivators [47,48,49]. Afshari and colleagues reported that it was more common that low exercisers lacked someone who could motivate them [15].

Moreover, from an intervention or applied perspective, one might argue that a significant change in coping styles reflects a “helpful” or “not so helpful” coping style in dealing with the individual situation along the course of disease in daily life. Within the relatively healthy “Low stable” group, even-minded optimistic reflections, such as “Humor”, “Wish for information” and “Ups and downs”, appear as particularly “helpful” to deal with the own stable situation of PD. For the “worsening” subgroup, it turned out that to “Compare with others” may not have been so helpful when PD increased in severity over time, whereas seeking for “Social support” was. Within the “High stable” subgroup, it was helpful to “Keep going” in a stable but unhealthy situation. In sum, this highlights the importance for social support as the disease progresses to stay active and be able to participate in society.

### 4.4. Limitations and Future Perspectives

As in all longitudinal studies, we had a drop-out that might induce a bias. We have previously reported that those that were lost to follow-up were significantly older and had a longer PD duration [50]. Nevertheless, our sample still represents the full span of disease severity (HY 1–5).

It also needs to be noted that the “Low stable” group was significantly younger than the other two groups, and we cannot exclude that some of the differences in coping styles between the groups are related to age. However, previous research that used the CPS in older adults without specifically having any chronic diseases found, inter alia, that the coping styles seeking “Social support” and “Comparison with others” were notably not significantly related to age [19]. Thus, our finding that these two styles change the most over time in the “Worsening” group is most likely related to the change in disease severity rather than to age differences or ageing. Moreover, there was no difference in age between the “Worsening” and the “High stable” groups, which would further support our interpretation of differentiated findings with respect to disease course.

In the current study, a change was considered present if the individual ratings exceeded one step on the five-point Likert scale. However, it needs to be noted that though this is a convention to capture true change, one could have made other decisions in this regard that may have better captured true change. We therefore highlight the need for studies that compare the psychometric properties of different coping instruments and verify their measurement errors in people with PD. Such studies are needed to verify whether a change exceeds the measurement error of the used instrument. The knowledge gained would be of value for future longitudinal studies and for intervention studies that use coping as an outcome. Future longitudinal studies of coping in people with PD might also benefit from collecting more descriptive data on treatments such as pharmacological, physical therapy and whether they had access to a multidisciplinary team.

## 5. Conclusions

The results revealed four distinct patterns of coping (pessimistic, optimistic, persistent, support-seeking) and suggest that even when facing severe challenges due to PD in daily life, coping styles remain relatively stable over time, which would support a dispositional theory of coping. However, and closer to the reality of PD progression over time, a differentiated perspective on stability versus worsening in PD severity appeared to trigger a contextual and process-related re-evaluation of coping styles, indicating the possibility of changing strategies when the familiar strategies prove to be less helpful. This result suggests the importance of supporting persons with PD by providing tools that facilitate adjustments to coping strategies when needed. Future studies should focus on finding causal relations between different coping styles and health outcomes.

## Figures and Tables

**Table 1 behavsci-10-00190-t001:** Descriptive information and participant characteristics at baseline (T1) and 3-year follow-up (T2), sample size N = 158.

Characteristic	T1 Baseline	T2 3-Year Follow-Up
Sex (men), *n* (%)	102 (64.6)	102 (64.6)
Age (years), mean (SD)	68.5 (8.6)	71.6 (8.7)
Education (≥11 years, i.e., high-school, university), *n* (%)	60 (38.0)	61 (38.6)
Living area (urban), *n* (%)	67 (42.4)	67 (42.4)
PD duration (years), median (q1–q3)	8 (5–12)	11 (8–15)
Disease severity (Hoehn and Yahr, HY, “on state”), *n* (%)		
HY I	35 (22.2)	10 (6.3)
HY II	52 (32.9)	36 (22.8)
HY III	37 (23.4)	66 (41.8)
HY IV	32 (20.3)	37 (23.4)
HY V	2 (1.3)	9 (5.7)
Cognitive function (MoCA), median (q1–q3)	26 (23–28) ^1^	26 (23–28) ^2^
Motor symptoms (UPDRS III), median (q1–q3)	29 (21–37) ^3^	27 (21–39) ^4^
Non-motor symptoms (NMS Quest), median (q1–q3)	10 (7–13) ^5^	11 (7–15) ^6^
Freezing of gait (FOGQsa), median (q1–q3)	6 (1–12) ^1^	8 (2–14) ^7^
Functional limitations/dependence on mobility devices, median (q1–q3)	4 (2–6)	5 (3–7)
Activities of daily living (PADLS) (q1–q3)	2 (2–2)	2 (2–3) ^3^

Note: In the table reported % is valid %. Higher scores = “worse” in all instances but MoCA, where higher scores are “better”. PD, Parkinson’s disease; MoCA, Montreal Cognitive Assessment (0–30); UPDRS (0–108), Unified PD Rating Scale, part III = motor examination; NMSQuest (0–30), Non-motor symptoms Questionnaire, FOGQsa, (0–24), self-administered Freezing of Gait Questionnaire; Functional limitations/dependence on mobility devices, rated by professional rater (0–14); PADLS (1–5), PD Activities of Daily Living Scale. ^1^ 2 missing; ^2^ 10 missing; ^3^ 1 missing; ^4^ 9 missing; ^5^ 23 missing; ^6^ 35 missing; ^7^ 3 missing.

**Table 2 behavsci-10-00190-t002:** Factor analysis of coping styles, using principal component analysis as extraction method and varimax rotation with Kaiser Normalization, sample size N = 158.

Coping Style ^1^	Factor 1	Factor 2	Factor 3	Factor 4
**Factor 1: Remaining faithful but often feeling blue (pessimistic)**				
Distraction	0.550	0.068	0.008	0.282
Faith	0.407	0.295	−0.160	−0.102
Giving up	0.589	−0.071	0.497	−0.022
Life loses meaning	0.740	0.008	−0.001	0.185
Comparison with others	0.580	0.198	−0.402	−0.148
Someone else to take over	0.663	−0.002	0.143	−0.069
**Factor 2: Being even-minded and optimistic despite ups and downs (optimistic)**				
Comparison with the past	0.137	0.531	−0.093	−0.080
Wish for information	−0.021	0.683	−0.156	0.226
Ups and downs	0.176	0.467	0.120	0.282
Humor	−0.033	0.761	0.073	−0.149
**Factor 3: Keep going and adapt to the situation (persistent)**				
Keep going	−0.111	0.306	−0.628	−0.022
Adaptation to the given	−0.113	0.389	0.638	−0.029
**Factor 4: Getting help and social support (support-seeking)**				
Social support	0.055	−0.001	−0.004	0.904

^1^ Coping style according to the Coping Patterns Schedule [19,30].

**Table 3 behavsci-10-00190-t003:** Stability of coping styles over time: T1 to T2, sample size N = 158.

			Rating of Coping Style	
Coping Style ^1^	Ranking of Stability	Stability ^2^ *n* (%)	T1 Mean (SD)	T2 Mean (SD)
Humor	1	143 (90.5)	3.6 (0.8)	**3.8 (0.8)**
Life loses meaning	2	141 (89.2)	2.0 (0.8)	1.9 (1.0)
Faith	3	138 (87.3)	2.4 (1.2)	2.3 (1.3)
Wish for information	3	138 (87.3)	**3.7 (0.9)**	**3.9 (0.9)**
Comparison with others	3	138 (87.3)	3.5 (0.9)	3.5 (1.1)
Adaptation to the given	6	137 (86.7)	**3.7 (0.7)**	3.7 (1.0)
Comparison with the past	7	136 (86.1)	**3.8 (0.8)**	3.7 (0.9)
Ups and downs	8	134 (84.8)	3.4 (0.9)	3.7 (0.9)
Social support	9	129 (81.6)	3.4 (0.9)	**3.8 (0.8)**
Someone else to take over	10	128 (81.0)	2.3 (1.0)	2.5 (1.2)
Giving up	11	125 (79.1)	2.2 (0.9)	2.4 (1.1)
Keep going	12	124 (78.5)	3.4 (1.1)	**3.8 (1.1)**
Distraction	13	119 (75.3)	3.3 (1.0)	3.4 (1.2)

^1^ Coping style according to the Coping Patterns Schedule [19,30]. For each coping style the respondent rates on a Likert scale ranging from 1 (= strongly disagree) to 5 (= strongly agree) how well it fits their own thinking and behavior. The three (at T2 three strategies share second place) coping styles rating highest on the scale (1–5) at each timepoint are bolded. ^2^ Stability was defined as a difference ≤1 in rating between T1 and T2 (i.e., on an individual level).

**Table 4 behavsci-10-00190-t004:** Comparison between Hoehn and Yahr (HY) subgroups, with regard to stability of coping styles, personal factors, functional limitations and cognitive functioning.

	HY Subgroup ^1^			
	“Low Stable” (*n* = 58)	“Worsening” (*n* = 29)	“High Stable” (*n* = 71)	*p*-Value ^2^
Variable	Median (q1–q3) Unless Stated Otherwise			
Number of stable coping styles,				
T1 to T2 ^3^	12 (11–12)	10 (9–12)	11 (10–12)	**0.006 ^4^**
Sex (men/women), %	69/31	72/28	58/42	0.257
Age at T1	66 (58–71)	71 (68–77)	70 (65–75)	**<0.001 ^5^**
Functional limitations and dependence of mobility devices,				
at T1	2 (1–3)	4 (3–5)	5 (3–7)	**<0.001 ^5^**
at T2	3 (2–5)	6 (4–8)	6 (4–8)	**<0.001 ^5^**
MoCA score at T1	28 (26–29)	26 (23–27)	24 (22–27)	**<0.001 ^5^**
MoCA score at T2	28 (26–29)	26 (22–27)	24 (21–27)	**<0.001 ^5^**

Note: Number of stable coping styles (0–13); Functional limitations and dependence on mobility devices (0–14); MoCA, the Montreal Cognitive Assessment (0–30), higher score = “better”. Significant differences are bolded. ^1^ “Low stable”: low (HY 1–2) at T1 and T2; “Worsening”: low (HY 1–2) at T1, high (HY 3–5) at T2; “High stable”: high (HY 3–5) at T1 and T2. ^2^ Mann–Whitney U test for ad hoc pairwise tests (if Kruskal–Wallis test between the three subgroups was significant), except chi-square test to compare of distribution of sex. ^3^ A coping style was defined as stable if the difference in rating between T1 and T2 (i.e., on an individual level) was ≤1 on the scale (1–5). ^4^ “Low stable” vs. “Worsening”, *p* = 0.002; “Low stable” vs. “High stable”, ns; “Worsening” vs. “High stable”, ns. ^5^ “Low stable” vs “Worsening”, *p* < 0.001; “Low stable” vs. “High stable”, *p* < 0.001; “Worsening” vs. “High stable”, ns.

**Table 5 behavsci-10-00190-t005:** Rating of coping styles over time, by Hoehn and Yahr (HY) subgroup; Factor solution revealed at baseline, in short: 1 = pessimistic, 2 = optimistic, 3 = persistent, 4 = support-seeking.

HY Subgroup “Low Stable”, i.e., Low (HY 1–2) at T1 and T2 (*n* = 58)					
Coping Style	Factor	T1 Mean (SD)	Coping Style	Factor	T2 Mean (SD)
**Adaptation to the given**	**3**	**3.8 (0.7)**	**Wish for information**	**2**	**4.0 (0.8)**
**Comparison with the past**	**2**	**3.8 (0.8)**	Humor	2	3.9 (0.8)
**Wish for information**	**2**	**3.7 (0.8)**	Ups and downs	2	3.9 (0.8)
Humor	2	3.6 (0.8)	Keep going	3	3.8 (1.0)
Keep going	3	3.5 (0.9)	**Adaptation to the given**	**3**	**3.7 (0.8)**
Ups and downs	2	3.5 (0.9)	**Comparison with the past**	**2**	**3.7 (1.0)**
Comparison with others	1	3.4 (1.0)	Social support	4	3.7 (0.8)
Social support	4	3.3 (0.9)	Comparison with others	1	3.5 (1.1)
Distraction	1	3.3 (1.0)	Distraction	1	3.4 (1.1)
Faith	1	2.4 (1.1)	Faith	1	2.4 (1.3)
Someone else to take over	1	2.2 (0.9)	Giving up	1	2.4 (1.0)
Giving up	1	2.2 (0.8)	Someone else to take over	1	2.2 (1.0)
Life loses meaning	1	1.7 (0.5)	Life loses meaning	1	1.7 (0.8)

Note: For each coping style, the respondents rate on a scale, with response options ranging from 1 (= strongly disagree) to 5 (= strongly agree), how well it fits their own thinking and behavior. Coping styles at top at T1 or T2 are bolded for both rating times.

**Table 6 behavsci-10-00190-t006:** Rating of coping styles over time, by Hoehn and Yahr (HY) subgroups; Factor solution revealed at baseline, in short: 1 = pessimistic, 2 = optimistic, 3 = persistent, 4 = support-seeking.

HY Subgroup “Worsening”, i.e., Low (HY 1–2) at T1, High (HY 3–5) at T2 (*n* = 29)					
Coping Style	Factor	T1 Mean (SD)	Coping Style	Factor	T2 Mean (SD)
**Wish for information**	**2**	**3.9 (1,0)**	**Social support**	**4**	**4.0 (0.6)**
Comparison with the past	2	3.8 (1.0)	**Wish for information**	**2**	**3.9 (1.0)**
Comparison with others	1	3.8 (0.7)	Comparison with the past	2	3.7 (1.1)
Adaptation to the given	3	3.8 (0.9)	Adaptation to the given	3	3.7 (1.0)
Humor	2	3.7 (0.9)	Ups and downs	2	3.7 (1.0)
**Social support**	**4**	**3.4 (0.9)**	Humor	2	3.6 (1.0)
Ups and downs	2	3.4 (0.8)	Distraction	1	3.4 (1.3)
Distraction	1	3.3 (1.1)	Comparison with others	1	3.3 (1.2)
Keep going	3	3.1 (1.3)	Keep going	3	3.3 (1.4)
Someone else to take over	1	2.5 (1.1)	Someone else to take over	1	2.9 (1.3)
Giving up	1	2.2 (1.0)	Giving up	1	2.6 (1.2)
Faith	1	2.2 (1.3)	Faith	1	2.0 (1.2)
Life loses meaning	1	2.0 (1.0)	Life loses meaning	1	2.0 (1.1)

Note: For each coping style, the respondents rate on a scale, with response options ranging from 1 (= strongly disagree) to 5 (= strongly agree), how well it fits their own thinking and behavior. Coping styles at top at T1 or T2 are bolded for both rating times.

**Table 7 behavsci-10-00190-t007:** Rating of coping styles over time, by Hoehn and Yahr (HY) subgroups; Factor solution revealed at baseline, in short: 1 = pessimistic, 2 = optimistic, 3 = persistent, 4 = support-seeking.

HY Subgroup “High Stable”, i.e., High (HY 3–5) at T1 and T2 (*n* = 71)					
Coping Style	Factor	T1 Mean (SD)	Coping Style	Factor	T2 Mean (SD)
**Comparison with the past**	**2**	**3.7 (0.8)**	**Keep going**	**3**	**4.0 (0.9)**
**Wish for information**	**2**	**3.7 (0.8)**	**Wish for information**	**2**	**3.8 (0.9)**
**Adaptation to the given**	**3**	**3.7 (0.7)**	Humor	2	3.8 (0.8)
Humor	2	3.6 (0.8)	**Comparison with the past**	**2**	**3.7 (0.8)**
Social support	4	3.6 (0.8)	Social support	4	3.7 (0.9)
Comparison with others	1	3.6 (0.9)	**Adaptation to the given**	**3**	**3.6 (1.0)**
**Keep going**	**3**	**3.5 (1.1)**	Ups and downs	2	3.5 (1.0)
Distraction	1	3.4 (1.0)	Comparison with others	1	3.5 (1.1)
Ups and downs	2	3.3 (1.0)	Distraction	1	3.5 (1.2)
Faith	1	2.4 (1.2)	Someone else to take over	1	2.5 (1.2)
Someone else to take over	1	2.4 (0.9)	Faith	1	2.4 (1.3)
Giving up	1	2.2 (0.9)	Giving up	1	2.4 (1.1)
Life loses meaning	1	2.1 (0.8)	Life loses meaning	1	2.1 (1.1)

Note: For each coping style, the respondents rate on a scale, with response options ranging from 1 (= strongly disagree) to 5 (= strongly agree), how well it fits their own thinking and behavior. Coping styles at top at T1 or T2 are bolded for both rating times.

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
