# Peer review of "Coping Styles among People with Parkinson’s Disease: A Three-Year Follow-Up Study"

_behavsci, 2020, doi:10.3390/bs10120190_

Round 1

Reviewer 1 Report

In this study, the authors have investigated how coping styles of patients with Parkinson’s disease evolve for three years. Coping behaviors were also analyzed in comparison to different courses of disease.
The paper is well written and conveys a clear message. Some minor revisions should improve the consistency of the manuscript.
In Table 4, the age of PD patients among the three groups reports significant differences between the low stable and worsening subgroup, as well as between low stable and high stable subgroup. How do the authors consider the differences in the age of the subgroups concerning the different coping styles? How the aging effects have induced modifications of coping style over time in the different subgroups? Could differences in the coping styles be biased due to the effect of aging?
In addition, the authors claim to collect data during the home visit when the participants declared to be at their best. How do the authors expect their OFF status to generate different results? What was the levodopa equivalent daily dose? How does the coping style change concerning LEDD over the three years?

Author Response

Reviewer #1

In this study, the authors have investigated how coping styles of patients with Parkinson’s disease evolve for three years. Coping behaviors were also analyzed in comparison to different courses of disease.

The paper is well written and conveys a clear message. Some minor revisions should improve the consistency of the manuscript.

In Table 4, the age of PD patients among the three groups reports significant differences between the low stable and worsening subgroup, as well as between low stable and high stable subgroup. How do the authors consider the differences in the age of the subgroups concerning the different coping styles? How the aging effects have induced modifications of coping style over time in the different subgroups? Could differences in the coping styles be biased due to the effect of aging?

Response: Thank you for this relevant comment. We acknowledge that we cannot exclude that some differences in coping styles could be related to age, though only with respect to the “Low stable” group versus the two others. We address this concern by adding the following paragraph to the Limitations in the discussion section (lines 367-375):

“It also needs to be noted that the “Low stable” group was significantly younger than the other two groups, and we cannot exclude that some of the differences in coping styles between the groups are related to age. However, previous research that used the CPS in older adults without specifically having any chronic diseases found, inter alia, that especially the coping styles seeking “Social support” and “Comparison with others” were not significantly related to age [19]. Thus, our finding that these two styles change the most over time in the “Worsening” group is most likely related to the change in disease severity rather than to age differences or ageing. Moreover, there was no difference in age between the “Worsening” and the “High stable” groups, which would further support our interpretation of differentiated findings with respect to disease course.”

In addition, the authors claim to collect data during the home visit when the participants declared to be at their best. How do the authors expect their OFF status to generate different results? What was the levodopa equivalent daily dose? How does the coping style change concerning LEDD over the three years?

Response: Thanks for the comments. To clarify why we collected data when participants were “on”, we added the following sentence in the methods section (lines 124-126):

"Conducting the assessments when the participants were ”off” would potentially make it difficult for the participants to adequately reflect and respond, which would thus render the data invalid.”

With regard to Levodopa, unfortunately we do not have access to the participants’ LEDD data. We agree that descriptive data on their treatments (e.g. pharmacological, physical therapy, access to a multidisciplinary team) would have been of interest. We have therefore added the following sentence in the discussion section on future perspectives (lines 383-385).

“Future longitudinal studies of coping in people with PD might also benefit from collecting more descriptive data on treatments such as pharmacological, physical therapy and whether they had access to a multidisciplinary team.“

Reviewer 2 Report

The manuscript " Coping styles among people with Parkinson’s disease: a three-year follow-up study" by Nilsson et al., studied coping styles in patients with Parkinson’s disease.

The manuscript is well written and was a pleasure to read. The authors gave a comprehensive background review and provided all the necessary details needed to understand the manuscript. The study presents novel findings that could be of interest to clinicians. No suggestions.

Author Response

Reviewer #2

The manuscript " Coping styles among people with Parkinson’s disease: a three-year follow-up study" by Nilsson et al., studied coping styles in patients with Parkinson’s disease.

The manuscript is well written and was a pleasure to read. The authors gave a comprehensive background review and provided all the necessary details needed to understand the manuscript. The study presents novel findings that could be of interest to clinicians. No suggestions

Response: We thank the reviewer for the positive assessment of our manuscript.